# Characterization of an mRNA-Encoded Antibody Against Henipavirus

**DOI:** 10.3390/cimb47070519

**Published:** 2025-07-04

**Authors:** Zixuan Liu, Bingjie Sun, Ting Fang, Xiaofan Zhao, Yi Ren, Zhenwei Song, Sijun He, Jianmin Li, Pengfei Fan, Changming Yu

**Affiliations:** Laboratory of Advanced Biotechnology, Beijing Institute of Biotechnology, Beijing 100071, China; 15733250930@163.com (Z.L.); kokosun@foxmail.com (B.S.); fangting1008vip@163.com (T.F.); xiaofan_zhao@foxmail.com (X.Z.); renyi1@zju.edu.cn (Y.R.); 15965724931@163.com (Z.S.); 18785398415@163.com (S.H.); lijmqz@126.com (J.L.)

**Keywords:** henipavirus, mRNA, UTR, antibody, mouse model, protection

## Abstract

Nipah and Hendra viruses are lethal zoonotic pathogens with no approved vaccines or therapeutics. mRNA produced via in vitro transcription enables endogenous protein expression and cost reduction. Here, we systematically screened natural and artificial untranslated regions (UTRs) and identified an optimal combination for expressing henipavirus-neutralizing antibody 1E5. We generated mRNA-1E5 encapsulated in lipid nanoparticles (mRNA-1E5-LNPs). In vitro, mRNA-1E5-LNPs achieved functional antibody expression levels of >1500 ng/mL. In BALB/c mice, intravenous administration of mRNA-1E5-LNPs induced rapid antibody elevation (peak at day 3), without hepatic toxicity or tissue inflammation. We established two Hendra pseudovirus models in biosafety level 2 facilities to evaluate the efficacy of mRNA-1E5-LNPs. Low-dose prophylactic administration effectively blocked entry of the Hendra pseudovirus. Notably, a single 0.5 mg/kg dose of mRNA-1E5-LNPs, stored at 4 °C for two months and administered 7 days prior, provided good protection. Our findings provide a therapeutic strategy for henipaviral infections and a blueprint for the development of mRNA-based antibodies against emerging viruses.

## 1. Introduction

Nipah virus (NiV) and Hendra virus (HeV) are negative-sense, single-stranded RNA viruses belonging to the genus *Henipavirus* within the family *Paramyxoviridae* [1]. Phylogenetic and genomic analyses have classified NiV into two main lineages: the Malaysian strain (NiV_MY_) and the Bangladeshi strain (NiV_BD_) [2]. Additionally, a variant of HeV (HeV g2) was recently identified in horses and flying foxes [3]. Hendra and Nipah viruses (HNVs) are highly pathogenic and cause severe respiratory diseases and encephalitis in various mammals, including horses, pigs, and humans, with mortality rates ranging from 40% to 75% [4]. The first known outbreak occurred in Malaysia in 1998, followed by outbreaks in Singapore, Australia, and Bangladesh [5,6]. In a recent outbreak in Kerala, India, six people were infected, resulting in two fatalities [7].

Surface glycoproteins, attachment glycoproteins (G), and fusion glycoproteins (F) of HNVs work in concert to mediate viral fusion with the host cell membrane. Both G and F are primary targets for neutralizing monoclonal antibodies [8]. Several neutralizing antibodies targeting either G or F have been reported and have demonstrated ideal protective efficacy in animal models [3,9,10,11]. Among these, m102.4, which targets the receptor-binding domain of G, has completed Phase I clinical trials, further confirming the potential of antibody therapy for HNV infections [12]. However, no approved vaccines or treatments are currently available for human use. Moreover, multiple novel HNV-like viruses have been discovered, expanding host species diversity and global distribution [13,14,15]. Previous studies have indicated that neutralizing antibodies may exhibit reduced or lost activity against these new isolates [16], presenting challenges for traditional antibody therapies.

With the rapid development of nucleic acid and lipid nanoparticle (LNP) delivery systems, messenger ribonucleic acid (mRNA)-based technologies have emerged as a promising approach for preventing and treating infectious diseases. In recent years, mRNA vaccines have transitioned from theoretical assumptions to practical applications, with the tremendous success of two mRNA vaccines during the COVID-19 pandemic [17,18]. In addition to vaccine development, preclinical studies on mRNA-based passive immunotherapy have made substantial progress in various fields, such as cancer, toxins, and infectious diseases [19]. Compared to recombinant proteins, mRNA-based antibody therapy offers potential advantages, including a faster response speed and a more straightforward production process. For instance, mRNA-1944, targeting the chikungunya virus, has completed Phase I clinical trials (NCT03829384) [20]. However, to date, no study has evaluated the potential application of mRNA antibodies in preventing HNV infection.

Immunogenicity is the primary limitation of applying therapy based on exogenous molecules. Improving the translation efficiency of mRNA molecules can reduce the dose and frequency of administration, making it a critical design approach to minimize immunogenicity. The 5′ and 3′ untranslated regions (UTRs) of mRNA are essential regulatory factors, with the 5′ UTR playing a more crucial role in translation efficiency [21,22]. In this study, we screened a range of natural, endogenous, and artificially designed UTRs to construct an optimal mRNA framework for the expression of the cross-neutralizing antibody 1E5, which targets the G protein [23]. We encapsulated mRNA-1E5 into LNPs, forming mRNA-1E5-LNPs, with in vitro expression levels exceeding 1500 ng/mL. In a mouse model, tail vein injection of mRNA-1E5-LNPs at the lowest dose (0.125 mg/kg) significantly inhibited rHIV-HeV g2. Notably, the administration of mRNA-1E5-LNPs seven days before the challenge still conferred robust protection against rHIV-HeV. This study highlights the potential of mRNA-1E5 as a prophylactic and therapeutic strategy against HNV infections, laying the groundwork for future research on antibody therapies utilizing mRNA.

## 2. Materials and Methods

### 2.1. Ethics Statement

All animal experiments were performed following the relevant guidelines of the National Regulations for the Administration of Laboratory Animals (Ethical Application No. IACUC-SWGCYJS-2022-001). Female BALB/c mice, aged 6–8 weeks, were housed in a suitable living environment with adequate water and free access to food.

### 2.2. Cells and Viruses

HEK293T cells were purchased from the American Type Culture Collection (ATCC, Manassas, VA, USA) and cultured in Dulbecco’s modified Eagle’s medium (DMEM) supplemented with 100 μg/mL streptomycin, 100 U/mL penicillin, and 10% fetal bovine serum (FBS) at 37 °C and 5% CO_2_. The experiments involving pseudoviruses were conducted in a biosafety level 2 facility.

### 2.3. Gene Construction

The full-length sequences of NiV_BD_ (GenBank: AY988601.1), NiV_MY_ (GenBank: FN869553.1), HeV (GenBank: NC_001906.3), HeV g2 (GenBank: MZ229748.1), GP, and FP used for pseudovirus packaging were synthesized and cloned into the pcDNA3.1 vector. The coding sequence of 1E5 H/L (Appendix A) was codon-optimized for Homo sapiens. The sequences necessary for in vitro transcription (IVT) of mRNA, containing the T7 promoter, 5′ UTR (Appendix A), 1E5 H/L chain coding sequence, 3′ UTR (Appendix A), and poly (A) tail (with/without), were cloned into the pUC57 vector by General Biotech (Chuzhou, China), with the restriction endonucleases *Eco*R I and *Hin*d III (New England Biolabs, Ipswich, MA, USA) at both ends.

### 2.4. mRNA Preparation and LNP Formulation

The IVT templates were linearized using *Hin*dIII-HF endonuclease (New England Biolabs) and extracted using a DNA fragment purification kit (Takara, Kusatsu, Japan). mRNAs were synthesized using the T7 High-Yield RNA Transcription Kit (Vazyme, Nanjing, China). The mRNA cap1 structure was added to the 5′ end using vaccinia virus Cap enzyme (Vazyme) and mRNA Cap 2′-O-methyltransferase (Vazyme). A poly (A) tail was added to the non-poly (A) transcripts using Escherichia coli poly (A) polymerase (Vazyme). The final mRNA products were purified, quantified, and stored at −80 °C until use.

High-purity mRNA was encapsulated in lipid nanoparticles (LNPs). The molar ratio of light to heavy chains was 1.1:1. The SM-102, DSPC, cholesterol, and mRNA (molar ratio of 50:10:38.5:1.5) were mixed with anhydrous ethanol. The encapsulation rate and concentration were measured using the Quant-iT RiboGreen RNA reagent and kit (Thermo Fisher Scientific, Waltham, MA, USA), and then the LNPs were stored at 4 °C.

### 2.5. In Vitro Expression of mRNA-1E5

The 293T cells were seeded in 24-well plates at a density of 3 × 10^5^ cells/well. When cultured to a confluency of ~80%, 1 µg of naked mRNA-1E5 H/L mixture and Lipofectamine 3000 were diluted in Opti-MEM I. The mixture was then incubated at room temperature for 15 min. Cells were transfected with the mixtures or directly added with 1 µg of mRNA-1E5-LNPs, and then cultivated at 37 °C under 5% CO_2_. The supernatant at each time point was collected, stored at −20 °C, and analyzed by ELISA and immunoblot analysis, as described below.

### 2.6. Western Blot

The supernatants of cells transfected with mRNA-1E5 H/L or 1E5 antibody were separated on a 4–12% SurePAGE gel (GenScript, Nanjing, China) under reducing or non-reducing conditions. Heavy- and light-chain detection was performed using goat anti-human IgG Fc-HRP (Abcam, Cambridgeshire, UK) diluted at 1:10,000 and mouse anti-human IgG kappa chain HRP (HuaBio, Hangzhou, China) diluted at 1:5000. Visualization was performed using the iBright 1500 gel imager system (Thermo Fisher Scientific). The target protein content was analyzed using ImageJ 1.8.0.

### 2.7. In Vivo Expression Dynamics of mRNA-1E5-LNPs

Female BALB/c mice (6–8 weeks old) were randomly divided into ten groups (*n* = 5/group). Animals were administered either mRNA-1E5-LNPs (1, 0.5, 0.25, or 0.125 mg/kg) or 1E5 antibody (5, 2.5, 1.25, or 0.625 mg/kg) via the intravenous route (i.v.) on day 0. Control groups received equal volumes (100 μL) of empty LNP particles or phosphate-buffered saline (PBS) per mouse. Changes in body weight were monitored from days 0 to 14. The orbital blood samples were collected at 0.25, 1, 3, 7, and 14 days after administration, and sera were collected after centrifugation at 4 °C for 10 min at 5000× *g*. The sera were then aliquoted and stored at −80 °C. Antibody concentrations were quantified by ELISA, as described below.

### 2.8. ELISA

The expression levels and dynamics of 1E5 were quantified using an ELISA assay. Microtiter plates (96-well) were coated with NiV_MY_ GP overnight at 4 °C. The coated plates were washed with PBST and blocked with PBS containing 2% (*w*/*v*) bovine serum albumin (BSA) (Sigma Aldrich, Saint Louis, MO, USA) at 37 °C for 1 h. The plates were then rewashed and incubated with serial dilutions of supernatants or mouse sera at 37 °C for 1 h, before three further washes and subsequent 1 h incubation with 100 µL of the horseradish peroxidase (HRP)-conjugated anti-human IgG antibody (Abcam) at a 1:10,000 dilution, followed by incubation with 100 µL of 3,3′,5,5′-tetramethylbenzidine (TMB) solution (Solarbio, Beijing, China) for 6 min at 25 °C. Then, 50 µL of termination buffer (Solarbio) was added to each plate. Absorbance was measured at 450/630 nm, and accurate quantification was conducted using a SpectraMax ABS microplate reader (Molecular Devices, Sunnyvale, CA, USA). Each quantitative test produced a standard curve using purified 1E5 to back-calculate the precise concentration of functional 1E5 molecules.

### 2.9. Pseudovirus Packaging and Neutralization Test

HEK293T cells were seeded into T75 flasks and prepared for transfection. The following day, the cells were transfected with a total of 18 µg of plasmid (15.4 µg of pNL4-3.Luc.R^-^E^-^, 1.3 µg of pcDNA3.1-HNV G, and 1.3 µg of pcDNA3.1-HNV F) with Lipofectamine 3000 transfection reagent (Invitrogen, San Diego, CA, USA) diluted in Opti-MEM I reduced serum medium. After 6 h post transfection, the medium was replaced with 18 mL of fresh medium, and the cells were then cultivated at 37 °C and 5% CO_2_ for an additional 42 h. The supernatant was collected by centrifugation at 4 °C and 800× *g* for 15 min, then filtered through a 0.45 μm syringe filter and stored at −80 °C.

HEK293T cells were plated in 96-well plates at a density of 3 × 10^4^ cells/well for the neutralization assays. When the confluency reached 80%, 50 μL of mRNA-1E5-LNP or 1E5 antibody was added in three-fold serial dilutions starting from 5 μg/mL. After 12 h of administration, 50 μL of HIV-pseudotyped NiV or HeV and 50 μL of medium were added to each well. Each plate contained six positive and six negative control wells. Only the pseudovirus was added to the positive control wells. Only isovolumetric medium was added to the negative control wells. After culturing at 37 °C and 5% CO_2_ for 48 h, 100 μL of medium was removed, and the cells were lysed with 100 μL of Bright-Lite™ luciferase assay reagent (Vazyme) for 2 min. A volume of 150 μL cell lysate was added to 96-well white assay plates, and the luminescence was measured using a GloMax Navigator microplate luminometer (Promega, Madison, WI, USA).

### 2.10. Challenge of BALB/C Mice with rHIV-HeV g2

Six-week-old female BALB/c mice were randomly divided into two groups (*n* = 3 for rHIV-HeV g2 and *n* = 2 for PBS). All animals were intraperitoneally (i.p.) injected with 500 μL rHIV-HeV-g2 or PBS. Bioluminescence imaging was performed daily for 1–14 days after challenge. Each mouse was anesthetized with isoflurane for 10 min and intraperitoneally injected with 150 mg/kg XenoLight D-Luciferin (PerkinElmer, Waltham, MA, USA). Bioluminescence signals and images were acquired using an IVIS Spectrum system (PerkinElmer), and quantitative analysis of the images was performed.

### 2.11. Prophylactic Efficacy of mRNA-1E5-LNP in Mice

The solution was administered through the tail vein 12 h in advance. Groups of 6-week-old female BALB/c mice (*n* = 40) were intravenously administered different doses of mRNA-1E5-LNPs (0.125, 0.25, 0.5, or 1 mg/kg) or an equal volume (100 μL) of PBS, which served as a placebo. After 12 h, each mouse was intraperitoneally injected with 500 μL of rHIV-HeV g2. Luminescence was detected following the steps described previously.

The drug was administered via the tail vein seven days in advance. The BALB/c mice were randomly divided into three groups (*n* = 5). Mice were administered a tail vein injection of 0.5 mg/kg mRNA-1E5-LNPs (stored at 4 °C for 8 weeks), 1E5 antibody, or an equal volume of PBS (100 μL). Seven days later, 500 μL of rHIV-HeV was injected intraperitoneally. The bioluminescence signals were measured in vivo.

### 2.12. Detection of Biochemical Criteria in Mice

Six-week-old female BALB/c mice were randomly divided into three groups (*n* = 5). All animals were intravenously administered either 0.5 mg/kg of mRNA-1E5-LNPs or an equal volume (100 μL) of PBS/LNPs. The serum was collected three days after administration. Following the manufacturer’s instructions, four biochemical enzyme levels—alanine aminotransferase (ALT), aspartate aminotransferase (AST), lactate dehydrogenase (LDH), and creatinine (Cr)—were quantified using corresponding assay kits (Beijing Beijian Xinchuangyuan Biotech, Beijing, China).

### 2.13. Histopathological Analysis in Mice

Four groups of six-week-old female BALB/c mice (*n* = 3) were randomly assigned. Each animal received an intravenous injection of either 0.5 mg/kg of mRNA-1E5-LNPs, 1E5 antibody, or an equivalent volume (100 μL) of PBS or LNPs. On day 3 post administration, tissues from the lungs, liver, spleen, kidneys, and heart were collected. These tissues were fixed in 10% formalin for 48 h, embedded in paraffin, sectioned, and stained with hematoxylin and eosin (H&E).

### 2.14. Statistical Analysis

All data analyses were performed using the GraphPad Prism 8.4.2 software. In all experiments, each data point represented the mean ± standard deviation (SD) of three replicates. Statistical significance was assessed using one-way ANOVA with multiple comparisons and analyzed using the Wilcoxon log-rank test. The unpaired *t*-test was used for comparisons between two groups. (ns, not significant; *, *p* < 0.05; **, *p* < 0.01; ***, *p* < 0.001; ****, *p* < 0.0001).

## 3. Results

### 3.1. Screening of Natural Endogenous UTRs

For effective protein biosynthesis, mature mRNA requires multiple components, including the 5′-terminal cap, UTRs flanking the coding sequence (CDS), and a 3′ polyadenylated (poly (A)) tail [24]. The poly (A) tail plays a crucial role in regulating mRNA stability, transport, translation, and degradation. Polyadenylation can be achieved through two principal strategies: template-directed synthesis during the IVT process using plasmid-incorporated poly (A) sequences [25], or post-transcriptional enzymatic addition through poly (A) polymerase activity [26]. Using the 5′ and 3′ UTRs from the human hemoglobin subunit alpha1 gene (*HBA1*), we constructed initial IVT templates for the 1E5 antibody, with or without a poly (A) tail, and inserted them into pUC57 vectors. A comparative analysis of the mRNA products generated by the two polyadenylation methods revealed distinct characteristics. The enzymatic approach demonstrated technical simplicity, achieving 3′ terminal extensions with approximately 120 adenylates within 60 min while maintaining transcript homogeneity (Figure 1A). Although enzymatic tailing yielded lower expression levels (462 ng/mL) than template-directed polyadenylation (867 ng/mL), the resulting titers remained sufficient for preliminary screening (Figure 1B). Considering the temporal constraints and technical complexity inherent in the synthesis and sequencing of poly (A)-containing plasmids, we implemented enzymatic tailing as a rapid screening approach for UTR optimization. This strategy facilitated the construction of IVT templates containing candidate UTRs, enabling the convenient evaluation of UTRs.

UTRs significantly affect mRNA translation, stability, and subcellular localization. Prior investigations on mRNA therapeutic platforms have predominantly employed endogenous UTRs derived from cellular genes [27,28]. In addition, the terminal oligopyrimidine (TOP) motif in UTRs suppresses translational activity [29]. Next, we compiled a collection of dozens of previously reported unused natural pairs of endogenous human UTRs. Eight candidate sequences were first selected for further evaluation. These UTR combinations were modified with or without the TOP motif to assess their effects on functional antibody expression. The UTRs from HBA1 and ribosomal protein S25 (Rps25) showed the highest translation efficiency. For Rps25 and Rps27a, removing the TOP motif resulted in a modest increase in translation efficiency. In contrast, for HBA1, the removal of the TOP motif led to a decrease in translation efficiency (Figure 1C).

### 3.2. Screening of Artificially Designed UTRs and Construction of mRNA-1E5

Integrating endogenous UTRs with artificially designed UTRs may represent a superior approach for UTR design [28]. The 5′ UTR and 3′ UTR exhibit distinct regulatory roles in modulating mRNA translation efficiency. The 5′ UTR directly governs translational output through its critical involvement in ribosome recruitment and translation initiation, whereas the 3′ UTR indirectly influences protein synthesis by modulating mRNA stability and degradation kinetics. Therefore, we sought to determine whether artificially designed 5′ UTRs could further enhance the translation efficiency of mRNA-1E5 when the 3′ UTR was held constant. Utilizing the PaddleHelix platform [30], we designed 20 distinct 5′ UTR sequences for the CDS of 1E5 heavy (D_H_1-20) and light (D_L_1-20) chains, respectively (Appendix A). The UTR lengths varied in 10-nucleotide increments from 20 to 50 nucleotides, with five sequences for each length. Subsequently, these 5′ UTRs were combined with the 3′ UTR of HBA1 to construct 1E5-H/L transcription templates. After IVT, we individually transfected mRNA-1E5 H/L into HEK293T cells. The supernatant was collected, and the expression of 1E5-H/L was detected by Western blotting. The results indicated that the translation efficiency of mRNA-1E5 H in D_H_1, D_H_2, D_H_3, D_H_5, and D_H_6 was higher than that in the natural HBA1 combination (Figure 2A). In contrast, the translation efficiency of mRNA-1E5 L was higher in HBA1 than in all D_L_1–D_L_20 UTR combinations (Figure 2B).

Next, we applied the high-efficiency 5′ UTRs identified for the heavy chain to the light chain. Specifically, we utilized D_H_1, D_H_2, D_H_3, D_H_5, and D_H_6 for both the heavy and light chains and compared their performances with those of two natural endogenous UTR combinations: HBA1 and Rps25. After individual transfection of mRNA-1E5 H/L into HEK293T cells, we assessed the expression of 1E5 H/L in cell supernatants by Western blotting (Figure 2C). Quantitative analysis using ImageJ revealed that heavy-chain expression was the highest in the D_H_1 combination, whereas light-chain expression was the highest in the Rps25 combination, with D_H_1 showing slightly lower expression (Figure 2D). These constructs were co-transfected into HEK293T cells, and the supernatant was collected to measure the expression of the complete 1E5 antibody using an ELISA assay. The results demonstrated that the functional antibody expression levels of the D_H_1, D_H_2, and D_H_3 combinations were significantly higher than those of the other candidate combinations, with D_H_1 exhibiting the highest expression (Figure 2E).

Subsequently, we synthesized plasmid templates of the D_H_1 and Moderna combinations with poly (A) tails to prepare complete mRNA via co-transcriptional tailing. We then compared the levels of antibody expression. The expression level in the D_H_1 combination was higher than that in the commercially used Moderna UTR combination (Figure 2F). Based on these results, we identified the D_H_1 combination as the most efficient UTR element for mRNA-1E5, and selected the D_H_1 UTR element for subsequent experiments. Thus, we established an efficient framework for the expression of mRNA-1E5.

### 3.3. In Vitro Efficiency Characterization of mRNA-1E5-LNP

We used a well-established encapsulation platform to prepare LNP-encapsulated mRNA-1E5 (mRNA-1E5-LNPs). The encapsulation efficiency of the mRNA-1E5-LNPs reached 93.8%, with a uniform particle size distribution and an average diameter of approximately 78.1 nm (Figure 3A). The in vitro expression level of the mRNA-1E5-LNPs was up to 1500 ng/mL, approximately three times higher than that of the same amount of naked mRNA-1E5 (Figure 3B). Western blotting confirmed the presence of complete IgG molecules in the expression supernatant after transfection with mRNA-1E5-LNPs (Figure 3C).

Next, we assessed whether the antibodies produced by mRNA-1E5-LNPs could exert a neutralizing effect, beginning with an evaluation at the cellular level. We used four pseudoviruses based on the recombinant human immunodeficiency virus (rHIV) skeleton, which carries the G and F proteins of HNVs. The mRNA-1E5-LNPs exhibited effective and broad neutralization 12 h before infection, with half-maximal inhibitory concentration (IC_50_) values ranging from 0.01 to 0.1 μg/mL, which were slightly lower than those observed in the IgG group (Figure 3D).

### 3.4. Preliminary Safety Evaluation and Expression Kinetics of mRNA-1E5-LNPs

To evaluate the potential hepatotoxicity of mRNA-1E5-LNPs in BALB/c mice, we assessed liver function by analyzing key biochemical indicators—ALT, LDH, AST, and CR—three days post administration. Our results revealed that mRNA-1E5-LNP administration did not significantly compromise liver function, with no statistically significant differences observed across all measured parameters (Figure 4A). Histopathological evaluation further demonstrated that, compared to the control group, the lung, liver, spleen, kidney, and heart tissues in the treatment groups exhibited no significant pathological changes, indicative of an absence of overt inflammation (Figure 4B).

To evaluate the expression kinetics of mRNA-1E5-LNPs, we administered 100 μL of mRNA-1E5-LNPs at various doses, 1E5 antibody, or an equal volume of PBS via tail vein injection into mice. Serum antibody levels and continuously monitored body weights were quantified at designated time points. All groups maintained relatively stable body weights, with fluctuations of less than 5% throughout the study. The absence of significant weight gain in the control group is likely related to the physiological stress induced by frequent blood sampling. No statistically substantial weight differences were observed between the mRNA-1E5-LNP and control groups (PBS/LNP) (*p* > 0.05; Figure 5A). Serum antibody concentrations in the mRNA-1E5-LNP groups exhibited rapid dose-dependent elevation, reaching 25.0 μg/mL (1 mg/kg) and 6.7 μg/mL (0.5 mg/kg) within 6 h post injection. Peak concentrations (42.3 μg/mL for the 1 mg/kg group and 16.3 μg/mL for the 0.5 mg/kg group, respectively) were observed on day 3, followed by a progressive decline to undetectable levels by day 14 (Figure 5B).

### 3.5. Establishment of a Mouse Evaluation Model Based on the HeV Pseudovirus

Given the BSL-4 containment requirement for handling authentic HNVs (category C pathogens), we employed pseudotyped virus technology to establish a BSL-2-compatible murine model to evaluate the protective efficacy of mRNA-1E5-LNPs. Building on prior findings demonstrating prophylactic and post-exposure protection of 1E5 against authentic NiV in hamsters [23], we focused on addressing the knowledge gap regarding HeV protection. We packaged the Hendra pseudovirus, whose surface could display natural F and G proteins. This approach utilized bioluminescent reporter genes to quantitatively monitor infection progression through in vivo imaging, as previously validated for the NiV pseudovirus [31]. Following intraperitoneal inoculation with high-titer rHIV-HeV g2, longitudinal monitoring detected bioluminescent signals as early as 48 h post infection (Figure 6A). Signal distribution analysis showed predominant abdominal localization, followed by thoracic accumulation, whereas PBS controls exhibited background-level intensity. The luminescence intensity peaked on day 6, followed by rapid clearance to undetectable levels by day 14 (Figure 6B). Based on this kinetic profile, we selected day 6 as the detection time for subsequent drug administration experiments for luminescence imaging.

### 3.6. In Vivo Prophylactic Protective Efficacy of mRNA-1E5-LNPs

Given the single-cycle replication nature of pseudoviral systems (where luciferase expression becomes irreversible post entry), we evaluated the protective efficacy of mRNA-1E5-LNPs against rHIV-HeV g2 using a pre-exposure prophylactic approach. A single intravenous dose of mRNA-1E5-LNPs (0.125–1 mg/kg) administered 12 h before the viral challenge demonstrated dose-dependent protection. Specifically, the high-dose groups (0.5 and 1 mg/kg) achieved near-complete viral suppression, with a reduction in luminescence intensity of greater than 95% compared to the PBS control (*p* < 0.0001). Even the lowest dose (0.125 mg/kg) maintained significant efficacy, with a 67.6% reduction in luminescence intensity (*p* < 0.0001) (Figure 7A,B).

At the cellular level, mRNA-1E5-LNPs exhibited relatively weaker neutralizing activity against rHIV-HeV than against other rHIV-HNVs (Figure 3D). To further assess the protective efficacy of mRNA-1E5-LNPs stored at 4 °C for over 8 weeks, we conducted a longer pre-exposure prophylaxis study in mice using rHIV-HeV. Tail vein injection of 0.5 mg/kg of mRNA-1E5-LNPs, purified 1E5 antibody, or an equal volume of PBS was administered seven days before intraperitoneal inoculation with rHIV-HeV. The results indicated that the luminescence intensity in the mRNA-1E5-LNP treatment group was significantly lower than that in the PBS control group (*p* < 0.0001) and the 1E5 antibody group (*p* = 0.03) (Figure 7C,D). This suggests that mRNA-1E5-LNPs provide more prolonged protection. These findings demonstrate the potent protective efficacy of mRNA-1E5-LNPs against pseudovirus infections.

## 4. Discussion

HNVs pose a significant threat to public health, owing to their high mortality rate and potential for outbreaks. Antibodies are essential for the antiviral response; however, the number of reported candidate antibodies against HNVs remains limited. Given the short incubation period and rapid progression of HNV infection, there is an urgent need for effective and rapid therapeutic interventions.

The field of mRNA therapeutics has advanced significantly over the past decade, emerging as a desirable platform for encoding and producing proteins of interest in vivo. By exploiting the natural role of mRNAs as transient carriers of genetic information for protein translation, mRNA-encoded antibodies offer several advantages over conventional approaches. First, rapid intracellular translation enables the swift production of antibodies, ensuring timely protection against acute infections. Animal studies indicate that mRNA antibodies achieve efficacy at lower doses than traditional methods, reducing therapeutic costs [32]. Additionally, the intrinsic properties of mRNA confer safety advantages by typically avoiding long-term immune activation, thereby minimizing adverse effects. However, critical challenges remain in optimizing dosage requirements and ensuring long-term safety. Current therapeutic applications require higher mRNA doses than prophylactic vaccines to maintain adequate concentrations, emphasizing the need for enhanced translational efficiency and refined safety profiles.

In this study, we explored the application of mRNA encoding a potent cross-neutralizing antibody, 1E5, which targets the G protein of HNVs. We optimized the untranslated elements of mRNA-1E5 by screening a series of naturally endogenous and artificially designed UTRs. The final optimized mRNA framework for mRNA-1E5 includes Cap1, a 5′ UTR (D_H_1), the CDS of the 1E5 antibody, a 3′ UTR (HBA1), and a poly (A) tail. This optimized mRNA framework was encapsulated in LNPs to form mRNA-1E5-LNPs, which demonstrated high in vitro expression levels and extensive neutralization against the four HNV pseudoviruses. Intravenous injection of mRNA-1E5-LNPs into BALB/c mice resulted in rapid and efficient expression of the 1E5 antibody, with no obvious hepatotoxicity and tissue inflammation. Serum antibody levels rapidly increased in all dosage groups, peaking on day 3 post administration.

Although the number of reported cases of HeV infection is lower than that of NiV infection, the potential for large-scale outbreaks cannot be ignored. Owing to the limitations of BSL-4 facilities, we established rHIV-HeV and rHIV-HeV g2 mouse models as alternatives for evaluating mRNA-1E5-LNPs and other relevant vaccines and antibodies. In Hendra pseudovirus mouse models, injections of mRNA-1E5-LNPs 12 h or 7 days before the viral challenge produced sufficient antibody concentrations to protect the mice from infection. Notably, mRNA-1E5-LNPs can be stored at 4 °C for 2 months without affecting their ability to express functional antibodies, and can significantly block pseudovirus infection, showing good stability and durable protective activity.

This study had several notable limitations. First, it lacked an in-depth analysis of the UTR sequences and structures that affect translation efficiency. Future work will explore optimal UTR combinations by integrating the length and secondary structure parameters to identify universal design principles for UTRs. Second, the lack of BSL-4 facilities and the non-replicative nature of the pseudovirus prevented the evaluation of the efficacy of mRNA-1E5-LNPs in live-virus models or post-exposure scenarios. Third, while the Phase I clinical trial of mRNA-1944 demonstrated favorable safety profiles at comparable doses [20], it must be acknowledged that the 0.5 mg/kg dosing regimen employed in this study does not represent an ideal low-dose therapeutic range for practical applications. Lastly, the LNP formulation requires further optimization to enhance translation efficiency, extend the short half-life of the mRNA antibodies, and mitigate potential safety concerns.

## 5. Conclusions

An efficient framework for mRNA-1E5 was established by optimizing its UTR components and comprehensively evaluating the expression levels and neutralizing activities of the 1E5 antibody both in vitro and in vivo. The established pseudovirus models offer a simple and effective preliminary screening approach for researching and developing therapeutic agents against the Hendra virus and other highly pathogenic pathogens. The optimized design of mRNA-1E5 highlights the application strategies and potential of mRNA technology in therapeutic antibodies against henipavirus, providing valuable insights and solutions for future research and development of mRNA-based therapeutics.

## Figures and Tables

**Figure 1 cimb-47-00519-f001:**
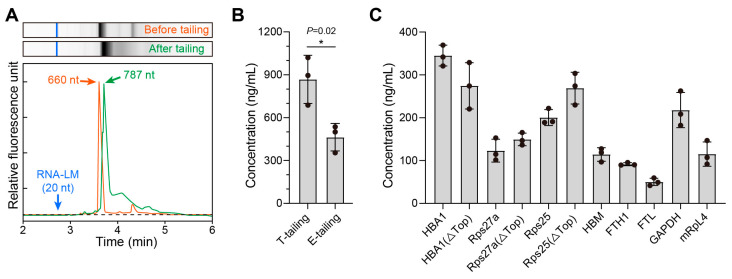
The establishment of UTR screening methods and the selection of natural endogenous UTRs. (**A**) Capillary electrophoresis analysis of mRNA-1E5 length and uniformity before and after enzymatic poly (A) tail addition. (**B**) Expression levels of mRNA-1E5 prepared by co-transcriptional poly (A) tail addition vs. enzymatic poly (A) tail addition in HEK293T cell supernatants (* *p* < 0.05). (**C**) In vitro expression levels of mRNA-1E5 with different natural endogenous UTRs.

**Figure 2 cimb-47-00519-f002:**
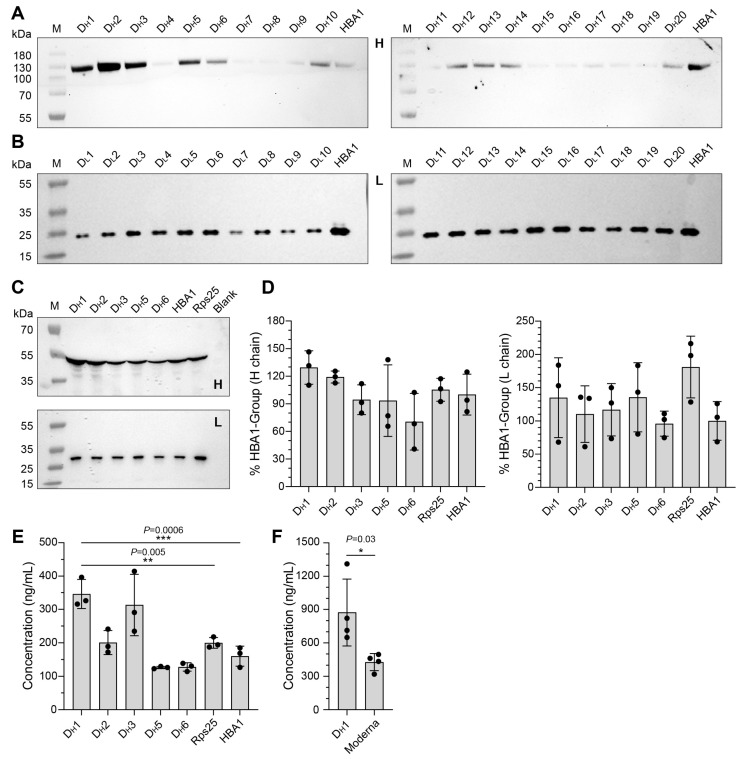
Screening of artificially designed UTR elements for mRNA-1E5 framework. (**A**,**B**) The expression levels of the mRNA-1E5 heavy (**A**) and light (**B**) chains using corresponding artificially designed UTRs in HEK293T cell supernatants after individual transfection. (**C**,**D**) The expression levels of the 1E5 heavy and light chains were determined in the cell supernatant collected after transfecting 293T cells with mRNA-1E5 H/L constructs. These constructs incorporated both artificially designed UTR combinations (D_H_1, D_H_2, D_H_3, D_H_5, D_H_6) and naturally endogenous UTR combinations (HBA1, Rps25). Panel (**D**) shows the relative gray value (% HBA1) of the immunoblot in Panel (**C**). (**E**) In vitro expression levels after co-transfection of mRNA-1E5 H/L. (**F**) Comparison of the mRNA expression levels of the D_H_1-HBA1 and Moderna UTR combinations (* *p* < 0.05; ** *p* < 0.01; *** *p* < 0.001).

**Figure 3 cimb-47-00519-f003:**
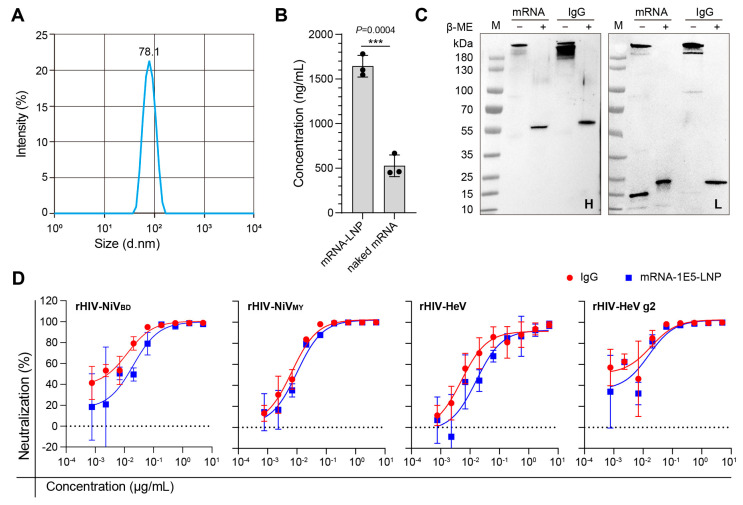
LNP encapsulation and neutralization capacity of mRNA-1E5. (**A**) The particle size of mRNA-1E5-LNPs. (**B**) Comparison of the expression levels of mRNA-1E5-LNPs with those of naked mRNA-1E5 (*** *p* < 0.001). (**C**) Detection of IgG and free light or heavy chains in the cell supernatant after transfection with mRNA-1E5-LNPs. Two images, from left to right, show the lanes: protein ladder (M), cell supernatant (non-reducing), cell supernatant (reducing), 1E5 antibody (non-reducing), and 1E5 antibody (reducing). (**D**) Neutralizing activity of IgG and mRNA-1E5-LNP against rHIV-HNV pseudoviruses.

**Figure 4 cimb-47-00519-f004:**
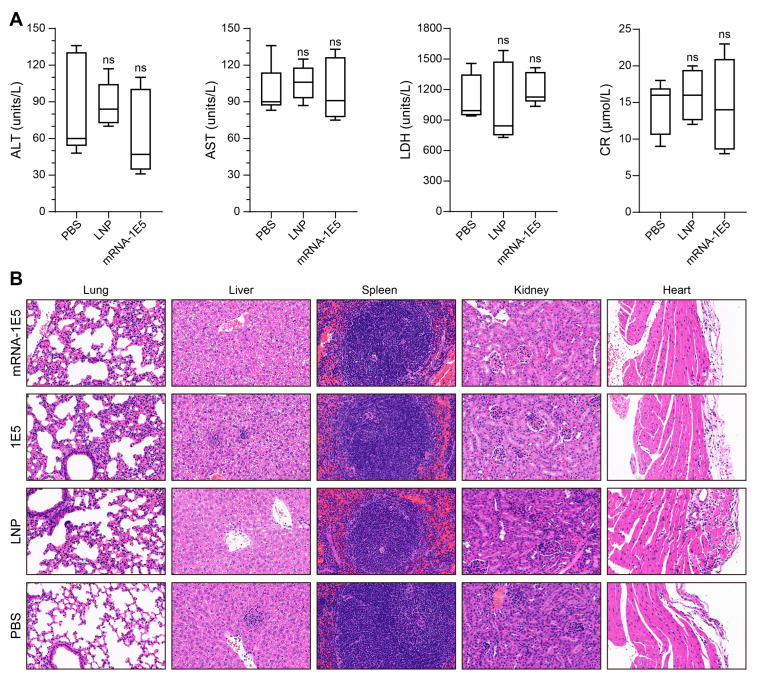
Hepatic toxicity and histopathological evaluation of mRNA-1E5-LNPs in mice. (**A**) Comparison of serum biochemical indicators (ALT: alanine aminotransferase; AST: aspartate aminotransferase; LDH: lactate dehydrogenase; CR: creatinine) 3 days post administration of mRNA-1E5-LNPs, PBS, or LNPs (*n* = 5, ns: not significant). (**B**) Histological examination of mouse tissues following treatment with the respective agents (*n* = 3). Representative images from three individual mice per group are displayed, with all images captured at identical magnification (40×) to ensure comparability.

**Figure 5 cimb-47-00519-f005:**
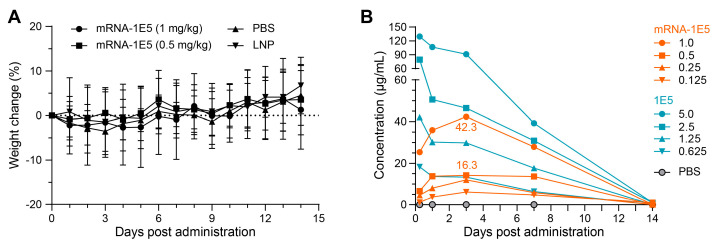
Changes in body weight (**A**) and functional 1E5 antibody concentrations (**B**) in mice within 14 days post administration of IgG or mRNA-1E5-LNPs (*n* = 5).

**Figure 6 cimb-47-00519-f006:**
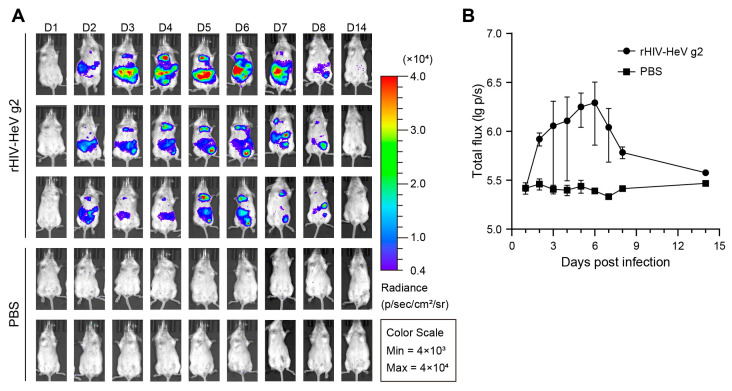
Establishment of a mouse model based on the HeV pseudovirus. (**A**,**B**) The expression of the luciferase reporter gene in BALB/c mice after intraperitoneal infection with rHIV-HeV g2 from day 1 to 14 (**A**), and differential analysis of the total luminescence intensity compared to the PBS group (**B**).

**Figure 7 cimb-47-00519-f007:**
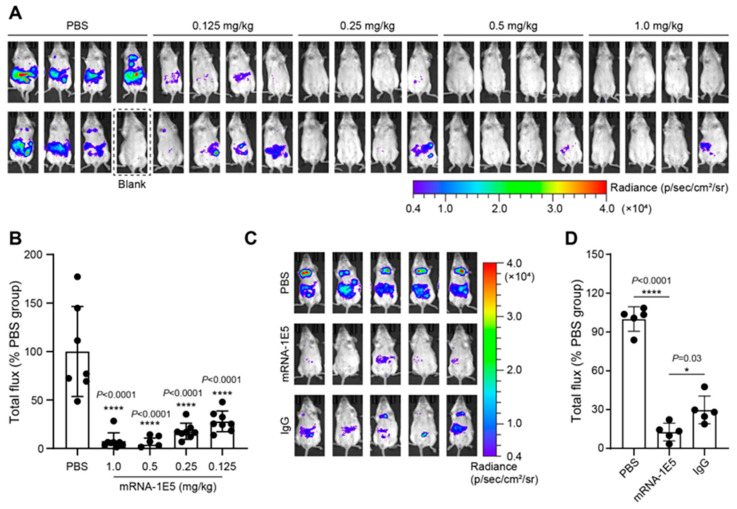
The prophylactic efficacy of mRNA-1E5-LNPs. (**A**,**B**) The protective effect of pre-treatment 12 h before on mice infected with rHIV-HeV g2 (*n* = 8). The luciferase expression was captured (**A**) compared to the total luminescence intensity of the PBS group (**B**) (**** *p* < 0.0001). (**C**,**D**) The protective effect of pre-treatment 7 days before the challenge against rHIV-HeV (*n* = 5). The expression of luciferase in the mice (**C**) was calculated and compared to that of the PBS group (**D**) (* *p* < 0.05; **** *p* < 0.0001).

## Data Availability

The authors declare that all the data supporting the findings of this study are available in the article and its Appendix A file.

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
