# Peer review of "Characterization of an mRNA-Encoded Antibody Against Henipavirus"

_cimb, 2025, doi:10.3390/cimb47070519_

Round 1
Reviewer 1 Report
Comments and Suggestions for Authors
This manuscript screened and identified an optimal 5′ UTR sequence that enables robust expression of an antibody 1E5 targeting the G protein of HNVs. In BALB/c mice model, the intravenous administration of mRNA-1E5-LNP rapidly increased antibody levels without hepatic toxicity. In Hendra pseudovirus models, low-dose prophylactic administration of mRNA-1E5-LNP could effectively block Hendra pseudovirus entry. Overall, the idea of this manuscript is clear, the manuscript is arranged reasonably and well-written. Some minor revisions are needed:
1. Line 181, it should be “mice”.
2. Now that the 3′ UTR of mRNA can modulate its stability and degradation kinetics, why was only the 5′ UTR optimized while the 3′ UTR was not?
3. Given that 1E5 targets the G protein of HNVs, why was the F protein included in pseudovirus?
Author Response
Comments 1: Line 181, it should be “mice”.
Response 1: Thanks for the correction. The error has been corrected.
Comments 2: Now that the 3′ UTR of mRNA can modulate its stability and degradation kinetics, why was only the 5′ UTR optimized while the 3′ UTR was not?
Response 2: Thank you for your insightful comment. Numerous studies have shown that the sequence features of the 5' UTR directly affect ribosome recruitment and are key for translation initiation and efficiency. After a thorough evaluation, we've prioritized optimizing the 5' UTR to achieve rapid and meaningful progress.
The 3' UTR has many regulatory elements, such as miRNA binding sites and AU-rich elements. They participate in complex RNA-protein interactions and regulate mRNA stability, degradation, and subcellular localization. However, their specific mechanisms and relationships vary a lot across different species, cell types, and physiological states, making things quite complex and diverse. We agree with your view on the 3' UTR's optimization potential and plan to refine it in future research.
Comments 3: Given that 1E5 targets the G protein of HNVs, why was the F protein included in pseudovirus?
Response 3: Thank you for your comment. The henipavirus membrane surface has two glycoproteins: the attachment glycoprotein (G) and the fusion glycoprotein (F). Both G and F are essential for viral entry into host cells. G binds to the host receptor Ephrin B2/B3, and then F's conformational changes drive viral-cellular membrane fusion. Pseudoviral particles must include both proteins to mimic this entry process and deliver reporter genes into target cells.
Reviewer 2 Report
Comments and Suggestions for Authors
The study focuses on the development of an mRNA-encoded antibody against henipaviruses, which is an area that has seen significant progress in recent years, particularly with the success of mRNA vaccines. While the study provides valuable insights into optimizing mRNA constructs for the expression of an antibody against henipaviruses, the limitations outlined above significantly weaken its overall impact and novelty. Detailed comments as follows.
1#The study uses one-way ANOVA with multiple comparisons and the Wilcoxon log-rank test for statistical analysis. While these methods are appropriate, the manuscript does not provide detailed information on sample sizes, power calculations, or adjustments for multiple comparisons.
2#The study mentions that mRNA-1E5-LNP did not significantly affect liver function in mice, but it does not provide comprehensive safety data, including long-term effects or potential off-target effects.
3#The study identifies an optimal UTR combination for mRNA-1E5 but does not compare its efficiency to other state-of-the-art mRNA constructs or delivery systems. This makes it difficult to assess the relative improvement offered by the optimized construct.
4#The manuscript includes several figures and tables, but some of the data presentations are complex and difficult to interpret. For example, the Western blot data (Figure 2C) and the ELISA data (Figure 3D and 3E) could benefit from clearer annotations and explanations.
5#There are many typos, including capital and small letters, mis-spellings, punctuation, and grammar issues.
6#The authors should acknowledged that 0.5mg/kg is not a low-dose level. Differences among experimental rodents, NHPs and humans should be well-discussed.
Author Response
Comments 1: The study uses one-way ANOVA with multiple comparisons and the Wilcoxon log-rank test for statistical analysis. While these methods are appropriate, the manuscript does not provide detailed information on sample sizes, power calculations, or adjustments for multiple comparisons.
Response 1: Thank you for your suggestion. We have added the relevant information in the manuscript.
Comments 2: The study mentions that mRNA-1E5-LNP did not significantly affect liver function in mice, but it does not provide comprehensive safety data, including long-term effects or potential off-target effects.
Response 2: Thank you for your insightful commentary. Safety profiles, particularly regarding inflammatory responses and long-term outcomes, are key concerns for current mRNA-based immunotherapies. Studies have also shown that the N1-methylpseudouridine modification, commonly used in mRNA vaccines, can induce ribosomal frameshifting during translation, potentially leading to off-target effects. We acknowledge that our preliminary research phase has limitations, which prevent the inclusion of more extensive safety data. We're dedicated to conducting deeper investigations into these aspects once mRNA antibody platforms undergo further optimization.
Comments 3: The study identifies an optimal UTR combination for mRNA-1E5 but does not compare its efficiency to other state-of-the-art mRNA constructs or delivery systems. This makes it difficult to assess the relative improvement offered by the optimized construct.
Response 3: Thank you for your commentary. In our previous study (PMID: 39560055), we assessed several UTR combinations from commercial and research sources. The natural HBA1 and Moderna's UTRs outperformed others in translation efficiency, so our analysis focused on them.
Though delivery platforms like exosome-based systems and non-ionic TNPs show potential in tissue targeting and reducing toxicity, LNPs are currently the most mature and widely used mRNA delivery technology. Our choice of LNPs was based on three key factors: their proven delivery efficiency across various cell types, well-documented safety in clinical use, and established scalability for large-scale therapeutic production.
Comments 4: The manuscript includes several figures and tables, but some of the data presentations are complex and difficult to interpret. For example, the Western blot data (Figure 2C) and the ELISA data (Figure 3D and 3E) could benefit from clearer annotations and explanations.
Response 4: Thanks for the suggestions. The annotations were added.
Comments 5: There are many typos, including capital and small letters, mis-spellings, punctuation, and grammar issues.
Response 5: Thank you for your meticulous attention to detail. We have rechecked the manuscript and corrected the typos.
Comments 6: The authors should acknowledged that 0.5mg/kg is not a low-dose level. Differences among experimental rodents, NHPs and humans should be well-discussed.
Response 6: Thank you for your feedback. While the Phase I clinical trial of mRNA-1944 demonstrated favorable safety profiles at equivalent doses (PMID: 34887572), it must be acknowledged that 0.5 mg/kg does not constitute a low-dose regimen, particularly given the unresolved long-term safety implications associated with higher mRNA payloads and lipid nanoparticle (LNP) components. We anticipate that iterative design optimizations, coupled with advancements in delivery vector engineering (e.g., tissue-specific LNPs or non-ionic TNPs), will enable substantial dose reduction in future therapeutic formulations.
Reviewer 3 Report
Comments and Suggestions for Authors
In the manuscript, Liu et al. conducted a comprehensive study to characterize an mRNA-encoded antibody against Henipavirus. The authors successfully identified an optimal UTR combination that enables robust expression of the antibody 1E5 by systematically screening a wide range of both natural and artificial UTR sequences. Additionally, they evaluated the expression levels of the generated mRNA-1E5-LNP and its efficacy against the virus in a mouse model. Overall, I believe the authors have made significant contributions in generating and evaluating an effective mRNA-based antibody targeting the Hendra virus both in vitro and in vivo. I have a few minor suggestions for the authors to consider:
- The authors mentioned that each experiment was performed with three replicates. For all bar plots presented in Figures 1 and 2, I suggest that the authors include individual data points in addition to the mean and standard deviation values to better represent the data distribution. Furthermore, it would be better to provide p-values for significant comparisons that support the key conclusions stated in the manuscript.
- In Figures 3D and 3E, the authors present the neutralizing activity of IgG and mRNA-1E5-LNP against the rHIV-HNVs pseudovirus. While the authors conclude that the IC50 values of mRNA-1E5-LNP are slightly lower than those of the IgG group, it may be challenging for the audience to discern the difference in neutralizing activity between these two groups, as the results are shown separately. I recommend that the authors consider a method to present these results side by side, which would facilitate easier comparisons and enhance the audience's understanding of the differences in neutralizing activity between mRNA-1E5-LNP and the IgG group.
Author Response
Comments 1: The authors mentioned that each experiment was performed with three replicates. For all bar plots presented in Figures 1 and 2, I suggest that the authors include individual data points in addition to the mean and standard deviation values to better represent the data distribution. Furthermore, it would be better to provide p-values for significant comparisons that support the key conclusions stated in the manuscript.
Response 1: Thank you for your suggestion. We have added individual data points and p-values in the figures.
Comments 2: In Figures 3D and 3E, the authors present the neutralizing activity of IgG and mRNA-1E5-LNP against the rHIV-HNVs pseudovirus. While the authors conclude that the IC50 values of mRNA-1E5-LNP are slightly lower than those of the IgG group, it may be challenging for the audience to discern the difference in neutralizing activity between these two groups, as the results are shown separately. I recommend that the authors consider a method to present these results side by side, which would facilitate easier comparisons and enhance the audience's understanding of the differences in neutralizing activity between mRNA-1E5-LNP and the IgG group.
Response 2: Thank you. Following your suggestion, we have now presented the relevant results in a side-by-side manner.
Round 2
Reviewer 1 Report
Comments and Suggestions for Authors
All my concerns have been addressed.
Author Response
Comments 1: All my concerns have been addressed.
Response 1: Thank you for your feedback.
Reviewer 2 Report
Comments and Suggestions for Authors
Authors addressed most of the comments well except the 2nd. Authors are required to perform preliminary safety evaluation such as HE staining at least. It will enhance the quality of study.
Author Response
Comments 1: Authors addressed most of the comments well except the 2nd. Authors are required to perform preliminary safety evaluation such as HE staining at least. It will enhance the quality of study.
Response 1: We appreciate your valuable comments. In accordance with your suggestion, supplemental histopathological evaluations were performed in mice to strengthen the preliminary safety assessment. The corresponding images and results are now included in the revised manuscript.
Round 3
Reviewer 2 Report
Comments and Suggestions for Authors
All the comments have been well-addressed.